# Brassinosteroid signaling-dependent root responses to prolonged elevated ambient temperature

Sara Martins[1], Alvaro Montiel-Jorda[1], Anne Cayrel[1], Stéphanie Huguet[2,3], Christine Paysant-Le Roux[2,3], Karin Ljung [4] & Grégory Vert [1]

Due to their sessile nature, plants have to cope with and adjust to their fluctuating environment. Temperature elevation stimulates the growth of *Arabidopsis* aerial parts. This process is mediated by increased biosynthesis of the growth-promoting hormone auxin. How plant roots respond to elevated ambient temperature is however still elusive. Here we present strong evidence that temperature elevation impinges on brassinosteroid hormone signaling to alter root growth. We show that elevated temperature leads to increased root elongation, independently of auxin or factors known to drive temperature-mediated shoot growth. We further demonstrate that brassinosteroid signaling regulates root responses to elevated ambient temperature. Increased growth temperature specifically impacts on the level of the brassinosteroid receptor BRI1 to downregulate brassinosteroid signaling and mediate root elongation. Our results establish that BRI1 integrates temperature and brassinosteroid signaling to regulate root growth upon long-term changes in environmental conditions associated with global warming.

---

[1] Institute for Integrative Biology of the Cell (I2BC), CNRS/CEA/Univ. Paris Sud, Université Paris-Saclay, 91198 Gif-sur-Yvette, France. [2] Institute of Plant Sciences Paris-Saclay IPS2, CNRS, INRA, Université Paris-Sud, Université Evry, Université Paris-Saclay, Bâtiment 630, 91405 Orsay, France. [3] Institute of Plant Sciences Paris-Saclay IPS2, Paris Diderot, Sorbonne Paris-Cité, Bâtiment 630, 91405 Orsay, France. [4] Umeå Plant Science Centre, Department of Forest Genetics and Plant Physiology, Swedish University of Agricultural Sciences, SE-901 83 Umeå, Sweden. Sara Martins and Alvaro Montiel-Jorda contributed equally to this work. Correspondence and requests for materials should be addressed to G.V. (email: Gregory.Vert@i2bc.paris-saclay.fr)

Human activities releasing billions of tons per year of carbon dioxide and other heat-trapping gases in the atmosphere change earth's climate with an unprecedented pace. According to the World Meteorological Organization (WMO, http://public.wmo.int), the earth's lower atmosphere temperature could rise of more than 4 °C by the end of the 21st century as a consequence of global warming. This gradual change in ambient temperature will impact all forms of terrestrial life, and especially plants that are rooted in soils and thus fixed to a specific location.

Temperature has a large impact on numerous plant developmental processes, including germination, growth, flowering, and hormonal responses, as well as on plant disease resistance[1, 2]. The bHLH transcription factor PHYTOCHROME INTERACTING FACTOR 4 (PIF4) has been shown to be central for multiple responses to warmer temperature in *Arabidopsis*, including flowering, nastic leaf movements, and elongation of the embryonic stem named hypocotyl[3–6]. In aerial plant parts, *PIF4* expression is under the control of the transcriptional repressor evening complex (EC) composed of EARLY FLOWERING 3 (ELF3), ELF4, and LUX ARRHYTHMO (LUX)[7–9]. Elevated temperature reduces the activity of the EC, promoting PIF4 accumulation. Warm temperature also increases the dark reversion at night of the PHYTOCHROME B (PHYB) thermosensor toward its inactive form, allowing PIF4 levels to build up[10, 11].

PIF4 directly binds to and activates the expression of the *YUCCA8* (*YUC8*) and TRYPTOPHAN AMINOTRANSFERASE OF ARABIDOPSIS 1 (*TAA1*) auxin biosynthetic genes upon elevated temperature to positively regulate auxin-dependent hypocotyl growth[3, 6, 12, 13].

The impact of ambient temperature elevation on underground organs is more elusive. Recent reports indicate that sudden increase of ambient growth temperature to 29 °C triggers auxin responses and accelerates root growth in an auxin-dependent manner[14, 15]. However, short-term changes in ambient growth temperature are largely buffered by the soil, indicating that roots are unlikely to face rapid changes in soil temperature. We therefore investigated the impact of prolonged elevation of ambient temperature, recapitulating the consequences of global warming. In the present study, we show that prolonged warmer conditions enhance root growth largely independently of auxin, by decreasing the levels of the brassinosteroid (BR) receptor BRASSINOSTEROID INSENSITIVE 1 (BRI1) to impact on BR-dependent root growth.

## Results

**Influence of temperature on root growth**. To evaluate the impact of prolonged elevated ambient temperature conditions on roots, we grew *Arabidopsis* seedlings at 21 and 26 °C and monitored primary root growth and developmental parameters. In our conditions, the

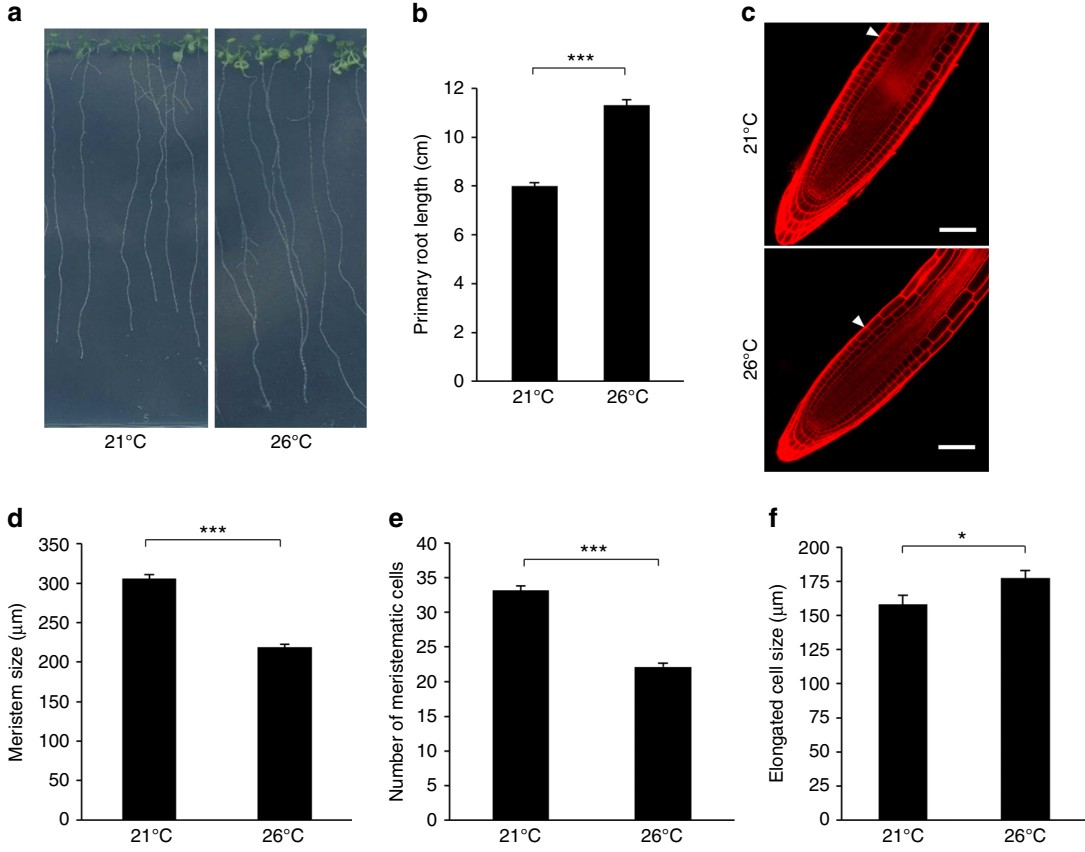

**Fig. 1** Elevated ambient temperature impacts on root elongation. **a** Phenotype of 15-day-old wild-type plants grown at 21 or 26 °C under long days. Representative pictures are shown. **b** Primary root length of 15-day-old wild-type plants grown at 21 or 26 °C (mean ± s.d., n = 25). The *asterisks* indicate a statistically significant difference (*t*-test, P < 0.0001). **c** Propidium iodide staining of wild-type plant roots grown as in **a**. *Arrows* represent the meristem boundary where cortical cells double their size. *Scale bar*, 50 μm. **d** Quantification of meristem size, defined by the distance from the quiescent center to the meristem boundary, in plants grown at 21 or 26 °C (mean ± s.d., n = 25). The *asterisks* indicate a statistically significant difference (*t*-test, P < 0.0001). **e** Quantification of meristem cell number in plants grown at 21 or 26 °C (mean ± s.d., n = 25). The *asterisks* indicate a statistically significant difference (*t*-test, P < 0.0001). **f** Quantification of differentiated root epidermal cell length (mean ± s.d., n = 25). The *asterisk* indicates a statistically significant difference (*t*-test, P < 0.01)

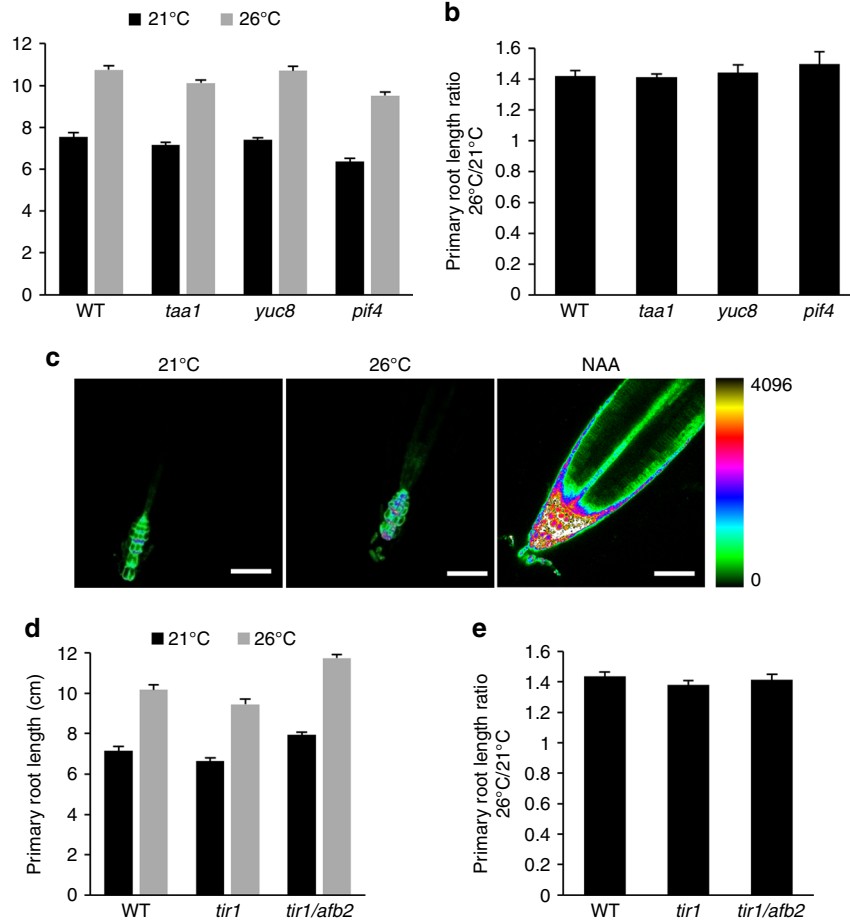

**Fig. 2** Elevated ambient temperature responses do not involve auxin. **a** Primary root length of 15-day-old wild-type plants (WT) and mutants impaired in hypocotyl temperature responses (mean ± s.d., n = 25). The difference between the different genotypes for a given growth temperature is not statistically significant. **b** Ratio of primary root length from plants grown at 26/21 °C as in **a** (mean ± s.d., n = 3). The response to elevated growth temperature between the different genotypes is not statistically significant. **c** Influence of elevated growth temperature on auxin responses using the DR5::GFP synthetic auxin reporter. Exogenous application of 1 μM NAA for 24 h is used as a control. Similar confocal detection settings were used to compare the three growth conditions. The look up table (LUT) allowing color coding of signal intensities is represented on the right. Representative images are shown. Scale bar, 50 μm. **d** Primary root length of WT and auxin perception mutants grown for 15 days at 21 or 26 °C (mean ± s.d., n = 25). The difference between WT plants, tir1, and tir1/afb2 mutants for a given temperature is not statistically significant. **e** Ratio of primary root length from plants grown at 26/21 °C as in **c** (mean ± s.d., n = 3). The response to elevated growth temperature between the different genotypes is not statistically significant

primary root grew significantly longer when plants were exposed to warmer temperature (Fig. 1a). Plants facing prolonged elevated ambient temperature showed a ~ 40% increase in root length compared to plants grown under standard conditions (Fig. 1b). Growth promotion results from increased cell size and/or cell number. The longer root phenotype of 26 °C-grown plants was not explained by a higher rate of cell division, as attested by the shorter meristem size, the reduced number of meristematic cells, and the decreased expression of the CYCB1;1 cyclin cell cycle marker gene (Fig. 1c–e; Supplementary Fig. 1a, b). To investigate if the root growth promotion observed at higher temperature resulted from increased root cell elongation, we measured the length of fully elongated root cells, using the first epidermal cell harboring a root hair bulge as a reference point. A significant increase in differentiated root cell length was found for plants grown at elevated temperature that, when cumulated, largely explains the longer root phenotype of 26 °C-grown plants (Fig. 1f). A 20 μm increase in cell size over the hundreds of elongated cells found in roots of 15-day-old plant indeed accounts for differences within the centimeter range. Taken together, these observations indicate that similar to short-term elevation of ambient temperature, constant warmth boosts primary root growth.

**Mechanism of temperature-mediated growth promotion.** In hypocotyls, PIF4 activates temperature-induced elongation through direct activation of the auxin biosynthetic genes YUC8 and TAA1 and increase in auxin levels[3, 6, 13]. To determine whether root responses to elevated temperature involve the same machinery as in hypocotyls, we first monitored primary root length of pif4, yuc8, and taa1 mutants grown at 21 and 26 °C. All three mutants showed longer roots at warmer temperature (Fig. 2a). The root growth promotion triggered by prolonged ambient temperature elevation was similar for wild-type, pif4, yuc8, and taa1, as highlighted by the ratio between primary root length at 26 and 21 °C (Fig. 2b). This indicates that long-term root responses to warmth do not require PIF4-, YUC8-, and TAA1-dependent auxin biosynthesis. PIF4 acts in concert with PIF5 in a number of biological processes including leaf senescence, phototropism, shade avoidance, and hypocotyl growth in response to low blue light[16–19]. We therefore tested whether PIF5 was acting redundantly with PIF4 in the temperature-mediated root elongation by characterizing the pif4/pif5 double mutant, but failed to observe any difference with wild-type plants (Supplementary Fig. 2a). Since PIF4/5, YUC8, and TAA1 are members of multigene families, our observations do not rule out the possible

involvement of other family members in temperature-mediated root elongation.

To further examine if auxin is involved in the differential root growth observed at 26 °C, we first evaluated the influence of growth temperature on the DR5::GFP auxin reporter. In contrast to the sharp increase in DR5::GFP expression observed following auxin treatment, prolonged ambient temperature elevation had little or no effect in DR5::GFP expression (Fig. 2c; Supplementary Fig. 2b). This was further supported by the wild-type primary root elongation response observed for the *tir1* and *tir1/afb2* auxin receptor mutants (Fig. 2d, e), although the same *tir1/afb2* double mutant fails to further elongate upon rapid temperature change from 22 to 29 °C[15]. The impact of higher-order auxin perception mutants could not be evaluated in our conditions since a large proportion of these plants arrest shortly after germination[20]. Altogether, this indicates that plants exposed to continuous elevated growth temperature elongate their primary root by a mechanism different than what has been described for hypocotyls and independently of auxin.

**Genome-wide root responses to elevated ambient temperature.** To shed light on the pathways mediating root growth responses to elevated ambient temperature, we performed whole-genome messenger RNA (mRNA)-seq analysis comparing root transcriptomes from plants grown at 21 and 26 °C. We found 2681 genes upregulated (false discovery rate < 0.05 and induction fold > 1.3) and 2114 genes downregulated (false discovery rate < 0.05 and repression fold > 1.3) by increased growth temperature, respectively (Supplementary Data 1). The top recurring gene ontology (GO) terms (ranked by false discovery rate < 0.05) among the upregulated genes at 26 °C were related to biotic stress and defense responses against bacteria, fungi, and herbivores (Fig. 3a; Supplementary Data 2). Elevated temperature therefore promotes root growth and likely primes root immune responses against potential threats that multiply vigorously with warmth. GO terms associated with genes downregulated with increased temperature include oxidative stress and responses to metal ions (Fig. 3a). This is likely caused by a massive deregulation of genes involved in iron homeostasis, with the nicotianamine synthase *NAS2* gene showing a 50-fold repression, and the iron uptake machinery genes *IRT1* and *FRO2* being downregulated 5- to 10-fold (Supplementary Data 3).

Whole seedlings exposed to a shift toward higher temperatures how strong over-representation of GO terms "response to auxin stimulus"[5]. Auxin-related terms were however not found among significantly enriched GO categories in our root-specific prolonged elevated temperature data set, which confirms that the genome-wide reprogramming of root gene expression upon elevated ambient temperature growth is largely independent of auxin responses. Several recent studies demonstrated the intricate role of BRs in root growth[21–24]. To unravel the mechanisms leading to longer roots upon growth at elevated temperature, we compared our list of genes regulated by elevated temperature in the root with a recent root-specific BR-regulated gene data set generated by RNA-seq[21]. Thousands of genes regulated by elevated temperature were also regulated by BRs in the root (Fig. 3b; Supplementary Data 4), and overlapped with genes directly bound by the BR signaling downstream transcription factor BRASSINAZOLE RESISTANT 1 (BZR1) in whole seedlings (Fig. 3c)[21, 25]. The root genomic responses to prolonged elevation in ambient temperature therefore have many facets, encompassing defense, nutrient homeostasis, and hormone responses that altogether drive the adjustment in root growth and physiology to warm conditions.

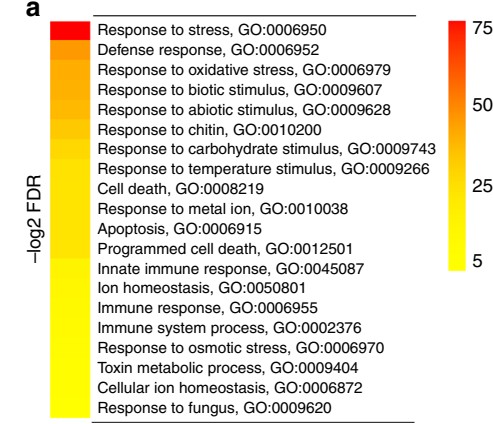

**a**

Response to stress, GO:0006950
Defense response, GO:0006952
Response to oxidative stress, GO:0006979
Response to biotic stimulus, GO:0009607
Response to abiotic stimulus, GO:0009628
Response to chitin, GO:0010200
Response to carbohydrate stimulus, GO:0009743
Response to temperature stimulus, GO:0009266
Cell death, GO:0008219
Response to metal ion, GO:0010038
Apoptosis, GO:0006915
Programmed cell death, GO:0012501
Innate immune response, GO:0045087
Ion homeostasis, GO:0050801
Immune response, GO:0006955
Immune system process, GO:0002376
Response to osmotic stress, GO:0006970
Toxin metabolic process, GO:0009404
Cellular ion homeostasis, GO:0006872
Response to fungus, GO:0009620

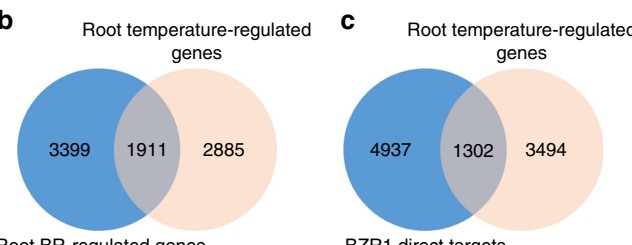

**Fig. 3** Genomic responses to prolonged elevated ambient temperature in roots. **a** Top 20 (ranked by false discovery rate (*FDR* < 0.05)) significantly enriched GO terms derived from genes that are regulated by increased temperature. The range of FDR is shown on the right. **b, c** Venn diagrams representing the overlap between elevated ambient temperature-regulated genes and published brassinosteroid-regulated genes in roots[21] **b**, or genes directly regulated by the binding of the BR-regulated downstream transcription factor BZR1[25] **c**

**Steroid signaling is involved in root responses to warmth.** BRs impact root growth by modulating the elongation of differentiated cells and meristem size[22, 23, 26]. As a result, the loss-of-function mutant for the *BRI1* gene encoding the BR receptor and the *bes1-D* constitutive BR response mutant for the *bri1* EMS SUPPRESSOR 1 (*BES1*) gene both show shorter roots and shorter meristems[22, 23]. To investigate the possible role of BRs in root responses to increased ambient temperature, we grew BR biosynthetic and BR signaling mutants with altered responses to BRs at 21 and 26 °C and scored their primary root length. The BR-deficient *det2* and *dwf4* mutants, carrying loss-of-function mutations in the *DEETIOLATED2* and *DWARF4* genes, respectively, displayed shorter root than wild-type plants at 21 °C (Supplementary Fig. 3a). Both mutants however showed wild-type temperature-mediated root elongation (Supplementary Fig. 3b), indicating that warm conditions interfere with root growth independently of BR biosynthesis. The *bri1* BR receptor mutant also showed shorter roots than wild-type plants at 21 °C (Fig. 4a), as previously reported[22, 23]. However, temperature elevation had a much greater impact on *bri1* roots (Fig. 4a), as visualized by the ~ 2-fold increase in primary root length ratio for *bri1* plants grown at 26 °C compared to 21 °C (Fig. 4b). This temperature-enhanced response of the *bri1* mutant is the dual consequence of increased root cell expansion and proliferation (Supplementary Fig. 3c, d).

To strengthen the genetic evidence supporting a role of BR signaling in temperature-mediated root growth, we scored root responses to warm conditions using genetic backgrounds showing enhanced BR responses. The *bes1-D* constitutive BR response mutant, which overaccumulates the dephosphorylated and active

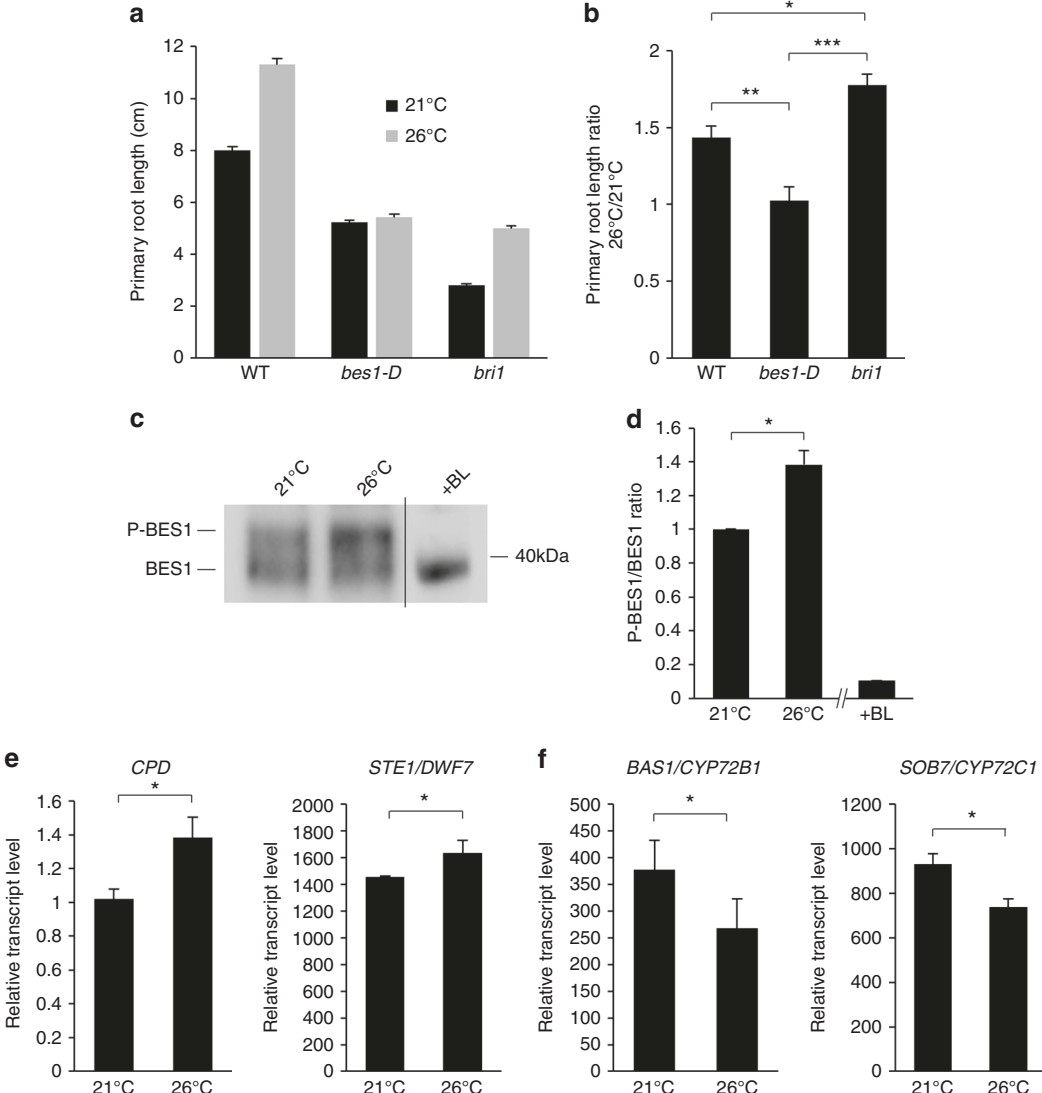

**Fig. 4** Brassinosteroid signaling regulates root responses to elevated ambient temperature. **a** Primary root length of 15-day-old wild-type (*WT*) or brassinosteroid-related mutant plants grown at 21 or 26 °C (mean ± s.d., *n* = 25). **b** Ratio of primary root length from plants grown at 26/21 °C as in **a** (mean ± s.d., *n* = 3). The *asterisks* indicate a statistically significant difference (Kruskal–Wallis; *$P < 0.01$; **$P < 0.001$; ***$P < 0.0001$). **c** Western blot analyses monitoring the phosphorylation state of BES1 in wild-type plants grown at 21 or 26 °C using anti-BES1 antibodies. A root protein extract from plants treated with brassinolide (*BL*) is used as a control for BES1 dephosphorylation. A representative blot is shown. **d** Ratio of phosphorylated BES1 (*P-BES1*) over BES1 levels (mean ± s.d., *n* = 4). The *asterisk* indicates a statistically significant difference (Mann–Whitney; *$P < 0.01$; **$P < 0.001$; ***$P < 0.0001$). **e** mRNA accumulation of the brassinosteroid feedback-regulated *CPD* and *STE1/DWF7* genes in roots from wild-type plants grown at 21 or 26 °C (mean ± s.d., *n* = 3). The *asterisk* indicates a statistically significant difference (Mann–Whitney, $P < 0.01$). **f** mRNA accumulation of the *BAS1/CYP72B1* and *SOB7/CYP72C1* brassinosteroid metabolism genes in roots from wild-type plants grown at 21 or 26 °C (mean ± s.d., *n* = 3). The *asterisk* indicates a statistically significant difference (Mann–Whitney, $P < 0.01$)

form of the BR-regulated transcription factor BES1[27], failed to respond to warm temperature under these conditions (Fig. 4a, b). The *bzr1-D* constitutive BR response mutant, which accumulates the active dephosphorylated form of BZR1[28], showed defective temperature-mediated root responses, although to a lesser extent than *bes1-D* (Supplementary Fig. 4a). Since BES1 and BZR1 integrate several intertwined pathways, we sought to investigate root temperature responses using BR hypersensitive genetic backgrounds targeting earlier steps in BR signaling. We therefore monitored primary root length of the *bin2/Atsk2-2/Atsk2-3* triple GSK3 mutant[29], defective in the negative regulation of the pathway downstream of BRI1. *bin2/Atsk2-2/Atsk2-3* showed reduced responses to elevated temperature, as attested by the lower primary root length ratio of plants grown at 26 vs. 21 °C

(Supplementary Fig. 4a). These genetic evidence clearly indicate that BR signaling negatively impacts on root responses to prolonged elevated growth temperature.

Since short-term elevation in ambient temperature from 22 to 29 °C has recently been shown to enhance root elongation in an auxin and TIR1/AFB2-dependent manner[15], we wanted to clarify further the role of auxin and assess the contribution of BR signaling in temperature-mediated root growth. Differences between the two studies may arise from (i) the requirement for BR signaling in responses to temperatures below 26 °C while auxin would drive responses to higher temperatures, (ii) the continuous vs. short-term elevated ambient temperature experimental setup used, or (iii) the influence of long day vs. short day photoperiod. First, to examine the possibility that BR signaling

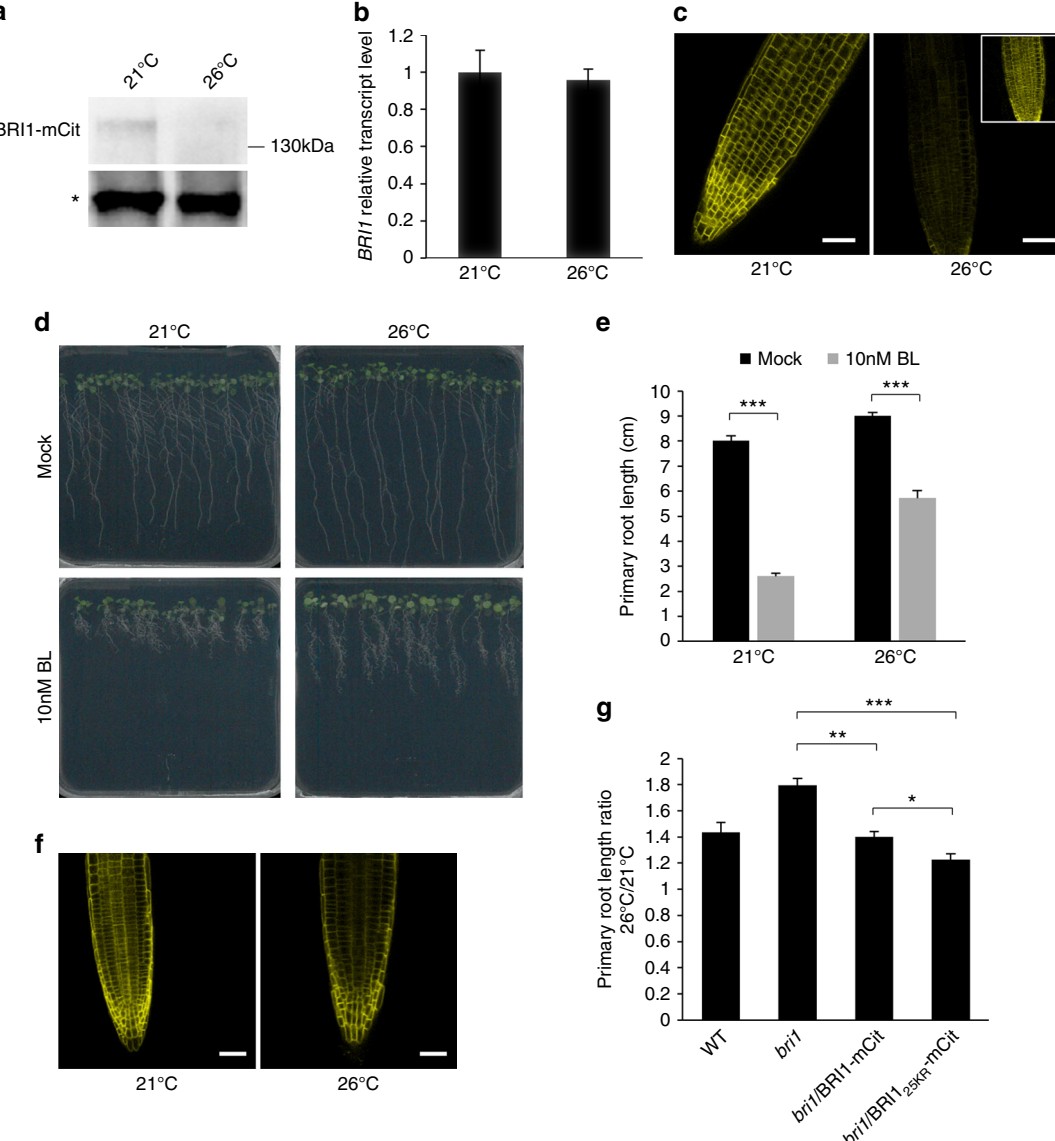

**Fig. 5** Elevated ambient temperature impinges on BRI1 levels and brassinosteroid perception. **a** Western blot analyses monitoring the accumulation of BRI1 protein in 15-day-old plants expressing a functional BRI1-mCitrine (mCit) fusion protein, under the control of the *BRI1* promoter, using anti-GFP antibodies. The *asterisk* represents a non-specific band used as loading control. A representative blot is shown. **b** Quantitative RT-PCR monitoring *BRI1* gene expression in roots from wild-type plants grown at 21 or 26 °C (mean ± s.d., $n = 3$). The difference in *BRI1* gene expression at 21 and 26 °C is not statistically significant. **c** Influence of elevated ambient temperature on BRI1-mCit protein accumulation in roots. Similar confocal detection settings were used to compare plants grown at 21 and 26 °C. Representative images are shown. Inset shows higher gain. Scale bar, 50 μm. **d** Phenotype of 15-day-old wild-type plants grown at 21 or 26 °C and treated with mock or 10 nM brassinolide (*BL*). Representative images are shown. **e** Primary root length of plants grown as in **d** (mean ± s.d., $n = 25$). The *asterisks* indicate a statistically significant difference (*t*-test, $P < 0.0001$). **f** Influence of increased growth temperature on BRI1-mCit and its non-ubiquitinatable BRI1$_{25KR}$-mCit variant. Similar confocal detection settings were used to compare the effect of temperature. Representative images are shown. *Scale bar*, 50 μm. **g** Ratio of primary root length from wild-type, *bri1*, *bri1*/BRI1-mCit, and *bri1*/BRI1$_{25KR}$-mCit plants grown at 21 and 26 °C for 15 days (mean ± s.d., $n = 4$). The *asterisks* indicate a statistically significant difference (Kruskal–Wallis; *$P < 0.01$; **$P < 0.001$; ***$P < 0.0001$)

operates only for responses to elevated temperatures up to 26 °C, wild-type, *bri1*, *bes1-D*, *tir1*, and *tir1/afb2* were subjected to continuous growth at 22 or 29 °C. In our conditions, wild-type plants showed similar root length at both temperatures, although hypocotyls dramatically elongated at 29 °C (Supplementary Fig. 4b). Continuous growth at 29 °C led to a sharp decrease in *bes1-D* primary root length, while *bri1* root elongated more at 29 °C than at 22 °C. Both *tir1* and *tir1/afb2* responded like wild-type plants to continuous growth at 29 °C. Second, to evaluate the role of BR signaling in root responses to a sudden increase in ambient temperature, we scored primary root length 4 days after transfer

from standard to elevated ambient temperature as previously reported[15]. Surprisingly, wild-type plants transferred to warmer conditions for 4 days failed to enhance primary root elongation compared to plants kept at 22 °C (Supplementary Fig. 4c), while hypocotyls were clearly longer. *bes1-D* showed shorter roots at 29 °C compared to 22 °C, whereas *bri1* still underwent promotion of root elongation at elevated temperature. Neither *tir1* nor *tir1/afb2* mutants were affected in their responses to temperature compared to wild-type plants. Finally, to assess the impact of photoperiod on the pathways underlying root growth at elevated temperature, we measured primary root length from plants

subjected to continuous growth at 21 or 26 °C under short days. Under short day conditions, primary root length of wild-type plants significantly showed a ~ 25% decrease with warmer temperature (Supplementary Fig. 4d). *bes1-D* roots were strongly shortened by continuous growth under elevated temperature, while *bri1* roots were less affected than wild type. Auxin perception mutants behaved like wild type under these conditions. Altogether, these different observations indicate that BR signaling modulates root growth in response to both short-term and prolonged elevated ambient temperature conditions, over a wide range of warm temperatures and independently of the photoperiod. Constant growth at elevated temperature has a profound influence on plant root growth depending on the photoperiod, with roots elongating more under long days and roots becoming shorter under short days. Our results indicate that BR signaling negatively regulates root responses to temperature elevation under long days, while positively regulating temperature-mediated root elongation under short days. Regardless, the primary root length ratio from plants grown at 26 °C over 21 °C are always greater for *bri1* and lower for *bes1-D* (Fig. 4b; Supplementary Fig. 4b–d). Interestingly, none of the conditions used in our study revealed a different behavior for auxin perception mutants in root temperature responses compared to wild-type plants, suggesting that additional factors are required to enhance the contribution to auxin in root growth responses to temperature.

Since our primary goal was to characterize root responses to ambient temperature elevation, especially in the context of global warming, we investigated deeper the influence of prolonged warmer temperature of only a few degrees (i.e., 21 vs. 26 °C) on BR signaling, using a long-day photoperiod. To this purpose, we first monitored the phosphorylation state of the BES1 transcription factor as a readout of BR signaling[30]. BES1 indeed accumulates under phosphorylated (P-BES1) and unphosphorylated (BES1) forms, and exogenous BR application promotes the conversion of BES1 to its unphosphorylated active form[27, 29]. Plants grown at 26 °C reproducibly accumulated more phosphorylated BES1 than plants grown at 21 °C (Fig. 4c), as evidenced by the increase in the P-BES1/BES1 ratio at elevated growth temperature (Fig. 4d). In addition, well-established BR homeostasis genes known to be regulated by BR signaling are affected by warm conditions. Expression of the *STE1/DWF7* and *CPD* BR biosynthetic genes, which are under a negative feedback regulation by BR signaling[31] and consequently induced in roots of *bri1* mutants compared to wild-type plants (Supplementary Fig. 5), is indeed higher at 26 °C compared to 21 °C (Fig. 4e). In contrast, the *BAS1* and *SOB7* genes involved in BR metabolism and induced by BRs[32, 33] are downregulated at elevated temperature (Fig. 4f). Taken together, these observations indicate that high temperature downregulates BR signaling to promote root elongation.

**Temperature impacts on BRI1 levels**. The mechanisms by which temperature impinges on BR signaling may be diverse. No significant changes in mRNA accumulation were observed in our whole-genome RNA-seq experiment for the major factors involved in BR signaling (e.g., *BRI1*, *BAK1*, *BES1*, *BZR1*) (Supplementary Fig. 6a). We therefore analyzed the levels of BRI1 protein in roots from plants grown at 21 and 26 °C. To this purpose, we took advantage of two isogenic transgenic lines expressing a functional fusion of BRI1 to the yellow fluorescent protein variant mCitrine (mCit) under the control of its own promoter, in the wild-type or *bri1* background[34]. In these lines, BRI1-mCit accumulates to the levels of endogenous BRI1 protein[35]. Interestingly, BRI1-mCit protein accumulated to lower

levels upon growth at elevated ambient temperature (Fig. 5a), while *BRI1* mRNA levels monitored by quantitative RT-PCR remained unaffected (Fig. 5b). This points to the existence of a post-transcriptional regulation of *BRI1* gene expression by elevated temperature. Confocal microscopy observations using similar detection settings for BRI1-mCit plants grown at 21 and 26 °C confirmed the decrease in BRI1-mCit protein levels at higher growth temperature (Fig. 5c). The post-transcriptional regulation of BRI1 by elevated temperature is also observed in *bri1*/35S::BRI1-mCit plants, where BRI1-mCit protein levels decrease upon warmth (Supplementary Fig. 6b). BRI1-mCit levels were however not affected in hypocotyls from seedlings grown at 26 °C compared to 21 °C, although the hypocotyl dramatically elongated with warmer temperature (Supplementary Fig. 7a). The influence of temperature on the levels of several plasma membrane proteins, including the plasma membrane intrinsic protein 2;1 aquaporin (PIP2;1) and the PIN-FORMED2 auxin efflux carrier (PIN2), was monitored to determine if the observed effect on BRI1 accumulation was specific. Neither PIP2;1 nor PIN2 levels were affected by elevated growth temperature showing that temperature specifically impacts on BRI1 levels to regulate root growth (Supplementary Fig. 7b, c). The effect of temperature on BRI1-mCit accumulation was also monitored with time after plants were transferred from 21 to 26 °C. The decrease of BRI1-mCit protein was first observed 12 h after plants were transferred to elevated ambient temperature growth conditions and was maximum after 36 h (Supplementary Fig. 7d, e), suggesting that this response is rather slow. Nevertheless, and consistent with the drop in BRI1 levels in the root, plants grown at 26 °C were hyposensitive to the root growth inhibitory effect of exogenously applied brassinolide (BL) (Fig. 5d, e).

We recently demonstrated that BRI1 undergoes lysine(K)-63 polyubiquitin-dependent endocytosis and degradation[35]. A functional but non-ubiquitinatable BRI1$_{25KR}$ mutant version is more stable at the cell surface and fails to reach the vacuole due to defective intracellular sorting[35]. To evaluate if BRI1 ubiquitination plays a role in the decrease of BRI1 protein levels and in the root responses to elevated ambient temperature, we subjected BRI1$_{25KR}$-mCit-expressing plants to growth at 21 or 26 °C. As observed by confocal microscopy analyses and in contrast to the decrease in BRI1-mCit levels observed at 26 °C (Fig. 5c), the non-ubiquitinatable BRI1$_{25KR}$ variant failed to respond to warmer growth conditions (Fig. 5f). This indicates that BRI1 ubiquitination mediates the drop in BRI1 levels at higher temperature. Consistently, transgenic *bri1* plants expressing BRI1$_{25KR}$-mCit under the control of its own promoter, which have been previously shown to be hypersensitive to BRs[35], displayed a diminished response to elevated temperature than plants expressing its wild-type BRI1-mCit counterpart (Fig. 5g). The primary root length observed for plants grown at 26 °C compared to 21 °C is ~40% greater for wild-type and *bri1*/BRI1-mCit control plants, while only ~ 20% longer for plants expressing the ubiquitination-defective BRI1 and within the range of what is observed for *bzr1-D* or *bin2/Atsk2-2/Atsk2-3* mutants (Supplementary Fig. 4a). The role of BRI1 ubiquitination in root temperature responses is underestimated due to the lower accumulation of BRI1$_{25KR}$-mCit protein than BRI1-mCit in these specific lines[35]. Altogether, these observations demonstrate that temperature impinges on BR signaling and BR-dependent root growth at the level of BRI1 ubiquitination.

## Discussion

Plant development is highly responsive to ambient temperature, and this trait has been linked to the ability of plants to adapt to

climate change[36]. The mechanisms of temperature perception and responses are slowly emerging, but are mostly limited to the influence of warm conditions on flowering or hypocotyl elongation.

Temperature elevation has long been known to trigger auxin production in shoots, and hypocotyl elongation[13]. Auxin has notably been proposed to mediate early phases of hypocotyl growth in response to temperature elevation, while steroid-dependent hypocotyl growth control drives later stages of the response to temperature[5]. Auxin also plays a role for hypocotyl elongation under continuous warmth[12], indicating that auxin and BRs may work together for long-term hypocotyl growth adjustment to changes in ambient temperature. Much less is known about how roots respond to prolonged ambient temperature elevation. In the present study, we demonstrated that root responses to elevated ambient temperature are associated with BR-dependent reprogramming of root growth. The mechanisms identified in this study that translate the temperature input into growth changes are radically different from aerial parts where auxin is the major driver of temperature-dependent growth[3, 6, 12, 13]. Although auxin levels build up upon temperature elevation in aerial parts, the contribution of shoot-derived auxin for root growth appears quite limited[37]. Unexpectedly, we demonstrated that prolonged elevated ambient temperature leads to a decrease in root auxin content (Supplementary Fig. 8a). Although reduced auxin levels may explain the accelerated root growth observed at 26 °C, we clearly proved that auxin played a minor role in responses to continuous growth at elevated temperature since tir1 and tir1/afb2 mutants behaved like wild type in our growth conditions. The auxin receptor TIR1 was shown to be stabilized in the root when plants are switched from standard to 29 °C growth temperature[15]. Growth under continuous elevated temperature leads to mild TIR1 accumulation (Supplementary Fig. 8b, c). This temperature-dependent stabilization of TIR1 likely compensates for the lower root auxin levels at 26 °C and explains the lack of differential auxin response. In our hands, the contribution of auxin to temperature-mediated root growth appears limited compared to the one of BR signaling. This suggests that additional factors are important to dictate the pathways driving root growth upon changes in ambient temperature.

Our work provides several complementary evidence using genetics, genomics, biochemical, and cell biology approaches that elevated temperature impinges on BR signaling at the level of BRI1 to control steroid signaling-dependent root elongation. Although genetic analyses clearly point to the negative role of BR signaling in root responses to elevated temperature in long days, our photoperiod of reference in this study, exogenous application of BL is associated with an enhanced response to temperature under the same conditions. Such discrepancy may be explained by the non-physiological concentration of exogenously applied ligand, which not only abolishes spatial and temporal control of BR responses that are crucial for proper root growth[21–23, 26, 38], but also likely activates other BR receptors such as BRL1 and BRL3 for which temperature regulation has not been demonstrated. Similarly, the fact that BR biosynthetic mutants show wild-type responses to warm conditions, while all insensitive or hypersensitive/constitutive BR genotypes tested respond differentially is surprising. This likely reveals some specificity in the cell type or developmental stage where BR are synthesized, perceived by BRI1 and BRLs, and where temperature acts on BR signaling.

BRs are mostly considered as a growth-promoting hormone. This is especially true for light grown seedlings where exogenous application of BL, or genotypes with enhanced BR responses, show longer hypocotyls. The fact that root elongation in responses to continuous warm conditions and long day

photoperiod is mediated by a decrease in BRI1 levels and attenuated BR signaling is therefore puzzling. In roots, BRs promote growth at very low concentrations and inhibit growth at higher levels, under standard growth temperature[22]. BR concentration may however be supraoptimal for root growth at 26 °C. Consequently, the temperature-dependent downregulation of BR signaling would yield longer roots. Two lines of evidence argue against this hypothesis. First, the bri1 mutant is still responsive to temperature elevation. Second, dose–response analyses to exogenously applied BL failed to confirm supraoptimal root BR levels since root growth was not inhibited at very low BR concentrations at 26 °C (Supplementary Fig. 9). A second explanation to the BR-mediated elongation upon heat may rely in the relative contribution of BRI1 and BR signaling to cell division and expansion. bri1 may mostly be affected in cell division under standard temperature, explaining the bri1 short root phenotype, while retaining the ability to promote cell elongation upon warmth. This scenario is however not supported by our observations that bri1 is severely impaired in both cell proliferation and elongation, and cannot easily explain why bri1 over-responds to temperature elevation compared to wild type. Alternatively, BR signaling may impinge on a downstream signaling pathway, specifically at elevated temperature, to differentially regulate root cell division and expansion. Downregulation of BRI1 by heat, or the absence of a functional BRI1, would unleash such downstream pathway and yield longer roots. This would explain the greater ratio of root length at elevated over standard temperature for bri1 compared to wild-type plants, regardless of the duration or the photoperiod, and the decreased ratio for bes1-D. We ruled out the involvement of auxin and TIR1/AFB2 in our growth conditions. Elevated temperature therefore likely influences root growth via another plant hormone, under the control of BR signaling. Gibberellins (GA) are known to modulate both the rate of cell proliferation and the extent of dividing and post-mitotic cell expansion[39–41]. Several recent reports in Arabidopsis and rice established a link between BR signaling and GA metabolism or GA signaling in the control of plant growth[42–46]. Although in most cases BRs positively regulate GA to boost growth, more complex interactions between the two hormones have been reported. BRs were notably found in rice roots to negatively regulate GA[45, 47]. BRs indeed promote the expression of the GA2ox3 GA-inactivating gene and inhibit the GA20ox3 GA biosynthetic gene in roots, leading to increased abundance of the central GA DELLA repressor SLR1[45, 47]. Decreased BR signaling has, therefore, the ability to yield more bioactive GA levels and enhanced GA signaling under certain circumstances or cell types. Future work will elucidate whether downregulation of BR signaling by temperature elevation promotes GA-dependent root growth.

The photoperiod strongly influences root growth and the role of BR signaling in root responses to elevated ambient temperature. While we have clearly demonstrated that warm conditions impact BR signaling by decreasing BRI1 levels in long days, the influence of short-day conditions on BRI1 levels and BR signaling will have to be investigated. The PIF-BES1/BZR1 module, which integrates BRs, temperature, and light inputs in hypocotyls[48, 49], may also contribute to root temperature responses and convey photoperiod input. It is therefore conceivable that temperature also crosstalks with BR and light signaling at the levels of BES1/BZR1 through interaction with a yet to be identified root-expressed PIF, providing more complexity in the mechanisms of ambient temperature control through BR signaling. We genetically excluded the contribution of PIF4 and PIF5, but other PIFs have already been reported to participate in root growth[50] and may therefore contribute to the integrated root growth control by BRs and temperature.

In nature, the rationale for plant roots growing deeper upon elevated ambient temperature appears unclear. Enhanced root growth may allow protection of the meristem bearing root stem cells from the superficial zone of the soil where the heat is more prominent. Alternatively, roots may grow deeper to search for water whose availability usually drops with warmth due to increased evaporation, and fewer precipitations under most climates. More importantly, our work offers the first glimpse on how climate change may impact plant underground organs and the whole plant. BRs are known to suppress immune signaling in plants through the activation of BZR1[51]. The increased expression in roots of a whole battery of defense genes against a wide variety of pests and pathogens upon elevated ambient temperature is therefore consistent with the observed downregulation of BR signaling. Setting up defense responses in the absence of threats is costly in energy, but may counteract the vigorous multiplication of bacteria, fungi, and insects upon elevated temperature. Surprisingly, the so-called growth trade-off between growth and immunity does not operate in the context of roots for warmer temperature since roots grow longer while activating immune responses[52]. Besides defense responses, iron homeostasis is also strongly affected by higher temperature. Considering the critical importance of iron for photosynthesis and its relatively low availability in soils, temperature-mediated changes in iron nutrition may have significant impact on plant yield. Although the molecular basis of elevated temperature responses is only emerging, it becomes increasingly important to obtain a clear picture of what the agriculture of tomorrow may look like to anticipate negative effects of temperature on plant physiology.

## Methods

**Plant materials and growth conditions**. The mutants and transgenic lines used in this study have been described previously: *pif4*, *pif4/pif5*[16], *taa1*[53], *det2*[54], *dwf4*[55], *bri1* knock-out[34], *bes1-D*[27], *bzr1-D*[28], *bin2/Atsk2-2/Atsk2-3*[29], *tir1*[56], *tir1/afb2*[20], *yuc8*[37], CYCB1;1::GUS, DR5:GUS, DR5::GFP[22], *bri1*/BRI1-mCit, BRI1-mCit[34], *bri1*/BRI1$_{25KR}$-mCit and BRI1$_{25KR}$-mCit[35]. Seeds were sterilized, stratified, and germinated on solid agar plates containing half-strength Linsmaier and Skoog medium without sucrose. After stratification, plates were incubated in growth chambers under long-day conditions (16 h light/8 h dark, 90 µE m$^{-2}$ s$^{-1}$) or short-day conditions (8 h light/16 h dark, 90 µE m$^{-2}$ s$^{-1}$), at 21, 22, 26, or 29 °C. For short-term elevated temperature experiments, plants were grown for 5 days at 22 °C before being transferred to either 22 or 29 °C for an additional 4 days. To ascertain that the BR- and auxin-related mutants used in this study were indeed affected in BR and auxin responses, respectively, we confirmed their sensitivity or resistance to BRs (or the biosynthetic inhibitor BRZ) and auxin using root length as a readout (Supplementary Fig. 10).

**Root growth parameters**. Quantitative measurements of primary root length were performed on scanned images of seedlings using the Fiji image software (http://fiji.sc/). The root meristem size was determined using propidium iodide. Briefly, seedlings were immersed in a solution of 10 µM propidium iodide for 10 min and then rinsed twice in distilled water before imaging. Meristem size was defined as the distance between the quiescent center and the meristem boundary where the cortical cell is twice the size of the immediately preceding cell. Cell divisions in root tips were visualized using plants expressing β-glucuronidase (GUS) under the control of the *CYCB1;1* promoter. GUS histochemical staining was performed using 5-bromo-4-chloro-3-indolyl-b-D-glucuronide as substrate[22]. The size of fully elongated differentiated cells was determined by measuring the length of the first epidermal cells showing a root hair bulge.

For all root growth parameters, 25 plants seedlings were used per treatment or genotype. Experiments were done in triplicates.

**Whole-genome RNA profiling**. Roots from plants grown at 21 or 26 °C for 10 days were collected and total RNA extracted using the RNEasy kit and DNase treatment according to the supplier's instructions (www.qiagen.com). RNA-seq experiment was carried out in triplicates at the POPS transcriptomic facility of the Institute of Plant Sciences Paris-Saclay, thanks to IG-CNS Illumina Hiseq2000 privileged access to perform paired-end (PE) 100 bp sequencing. RNA-seq libraries were prepared according to the TruSeq_RNA_SamplePrep_v2_Guide_15026495_C protocol (www.illumina.com). The RNA-seq samples were sequenced in paired-end (PE), with a sizing of 260 bp and a read length of 100 bases. Six samples per lane of Hiseq2000 were sequenced, using individual bar-coded adapters, and yielded ~25 millions of PE reads per sample.

**Bioinformatic analyses**. To facilitate comparisons, each RNA-Seq sample followed the same pipeline from trimming to count of transcript abundance. Read preprocessing criteria included trimming library adapters and performing quality control checks using FastQC (Version 0.10.1). The raw data (fastq) were trimmed for Phred Quality Score > 20, read length > 30 bases and sort by Trimming Modified homemade fastx_toolkit-0.0.13.2 software for ribosomal RNA sequences. Bowtie version 2 was used to align reads against the *Arabidopsis thaliana* transcriptome[57]. The 33602 mRNAs were extracted from TAIRv10 database with one isoform per gene to avoid multi-hits[58]. The abundance of mRNAs was calculated by a local script, which parses SAM files and counts only PE reads for which both reads map unambiguously to the same gene. According to these rules, around 98% of PE reads aligned to transcripts for each sample. Gene expression was analyzed using the EdgeR package (Version 1.12.0)[59], in the statistical software "R" (Version 2.15.0)[60]. GO analyses were performed using the AgriGO portal[61].

**Auxin content**. For quantification of free IAA, wild-type plants were grown at 21 or 26 °C for 15 days. Roots were collected, weighed, and root samples containing 10–25 mg root material (fresh weight) were extracted and purified as described[62]. IAA was quantified using GC-MS/MS (gas chromatography-tandem mass spectrometry)[62], using eight biological replicates.

**Western blot**. Protein extraction was conducted using roots from 15-day-old seedlings. Detection of BES1 and BRI1-mCit protein levels used the anti-BES1[63] (dilution 1/5000) and anti-GFP (Miltenyi Biotech 130-091-833, dilution 1/5000) antibodies, respectively. Western blots were done in triplicates. Scans of full blots used in this study are available (Supplementary Fig. 11).

**Quantitative real-time PCR**. Total RNA was extracted from roots using the GeneJet Plant RNA purification kit (ThermoFischer, www.thermofischer.com). First-strand complimentary DNA (cDNA) was synthesized from 2 µg of total RNA using the RevertAid Reverse Transcriptase (ThermoFischer, www.thermofischer.com). Primer design was carried out using the Primer3 software (http://biotools.umassmed.edu/bioapps/primer3_www.cgi). Primer combinations showing a minimum amplification efficiency of 90% were used in real-time RT-PCR experiments (Supplementary Table 1). For transcript normalization, we used the well-established At1g13320 reference gene encoding a protein phosphatase 2A subunit[64]. Real-time RT-PCR amplifications were performed using the Light Cycler Fast Start DNA Master SYBR Green I kit on a Light Cycler apparatus according to manufacturer's instructions (Roche, www.roche.com). Cycling conditions were as follows: 95 °C for 10 min, 40 cycles at 95 °C for 15 s, 60 °C for 15 s, and 72 °C for 15 s. PCR amplification specificity was checked using a dissociation curve (55–95 °C). A negative control without cDNA template was included for each primer combination. Experiments included three technical replicates and three independent biological replicates. Ratios were done with constitutive controls for gene expression to normalize the data between different biological conditions.

**Confocal microscopy**. Plant samples were mounted in water and viewed on a Leica TCS SP2 SP8-X confocal laser scanning microscopes (www.leica-microsystems.com/home/). For imaging GFP and propidium iodide, the 488-laser line was used. Excitation of mCit and Venus used the 514-nm laser line. Laser intensity and detection settings were kept constant in individual sets of experiments to allow for direct comparison of protein levels. Quantification of total fluorescence intensity in roots was performed using the Leica Confocal Software.

**Statistical analyses**. Statistical significance of the biological parameters ($n = 25$) was assessed using *t*-test (two genotypes/conditions) or the one-way analysis of variance (three genotypes/conditions and more) (*$P < 0.01$; **$P < 0.001$; ***$P < 0.0001$). Quantification of western blots, quantitative PCR, primary root length ratios, and auxin content experiments used the non-parametric Mann–Whitney (two genotypes/conditions) or Kruskal–Wallis (three genotypes/conditions and more) tests (*$P < 0.01$; **$P < 0.001$; ***$P < 0.0001$). Statistical analyses were performed with the software GraphPad Prism.

**Data availability**. RNA-seq data were deposited at Gene Expression Omnibus (accession #GSE99283; www.ncbi.nlm.nih.gov/geo/) and at CATdb databases (Project: NGS_2015_05_HIGH_TEMPERATURE_ROOT; http://urgv.evry.inra.fr/CATdb/) according to the "Minimum Information About a Microarray Experiment" standards. The authors declare that all other data supporting the findings of this study are available within the manuscript and its supplementary files are available from the corresponding author upon request.

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

## Acknowledgements

We are grateful to Yvon Jaillais and Ullas Pedmale for helpful discussions. We thank Ullas Pedmale, Joanne Chory, Mark Estelle, Yunde Zhao, Yvon Jaillais, and Ana Cano-Delgado, and for sharing *pif4*, *pif4/pif5*, *taa1*, *tir1*, *tir1/afb2*, TIR1-Venus, *yuc8*, *bri1/35S::BRI1-mCit*, CYCB1;1::GUS, DR5:GUS, and DR5::GFP seeds. We also thank Roger Granbom for skillful technical assistance. The present work has benefited from the I2BC Imagerie-Gif imaging facility, member of IBiSA supported by "France-BioImaging" (ANR-10-INBS-04-01) and the Labex "Saclay Plant Science" (ANR-11-IDEX-0003-02). The POPS transcriptomic facility is supported by the LabEx Saclay Plant Sciences (ANR-10-LABX-0040-SPS). S.M. is sponsored by a PhD fellowship from the Saclay Plant Sciences LabEx initiative (ANR-10-LABX-0040-SPS) funded by the French government and the Agence Nationale de la Recherche (ANR-11-IDEX-0003-02). K.L. is supported by the Swedish Governmental Agency for Innovation Systems (VINNOVA), the Swedish Research Council (VR), and Kempestiftelserna. This work was supported by grants from CNRS (ATIP to G.V.), Marie Curie Action (PCIG-GA-2012-334021 to G.V.), and Agence Nationale de la Recherche (ANR-13-JSV2-0004-01 to G.V.).

## Author contributions

S.M., A.M-J., and G.V. designed the experimental strategy and analyzed data. A.C. performed q-PCR analyses. K.L. measured auxin content. S.H. and C.P.-L.R. performed the RNA-seq experiments. G.V. wrote the manuscript. All authors commented on the manuscript.

## Additional information

**Competing interests:** The authors declare no competing financial interests.

