## [Peer Review File · Nature Communications]

Reviewers' comments:

Reviewer #1 (Remarks to the Author):

High temperature affects plant growth. In Arabidopsis, it induces hypocotyl elongation via different signaling components including those regulated by auxin and BR. A recent study (Wang et al, Nat Commun. 2016) has proposed that high temperature also induces Arabidopsis root elongation, and this response depends on auxin signaling.

The manuscript by Martins et al demonstrates that high temperature increases root elongation independent of auxin. Instead, it interacts with the BR signaling. A mutant in the BR receptor (*bri1*) is more sensitive to high temperature (shown as increased ratio of root length at 26°C over 21°C as compared to wild type). In agreement, a mutant with a constitutive BR response (*bes1-D*) is insensitive (ratio is 1). Comparative transcriptome analysis shows overlap between genes regulated by high temperature and genes regulated by BR. No overlap is observed between genes regulated by high temperature and genes regulated by auxin and shade avoidance response. Interestingly, high temperature reduces the accumulation of BRI1 in the plasma membrane.

The interaction between high temperature and BR signaling in the root is a very interesting finding. However, a major concern is the limited data provided by the authors to support their main conclusions according to which BR signaling represses root response to high temperature and the importance of high temperature mediated destabilization of BRI1. The comments below should assist the authors to better convey the main message of this work.

The effect of high temperature on BRI1 (Figure 4) is dramatic, by 36 hours no BRI1 protein is detected (extended Data Figure 4). It is therefore puzzling that after 15 days of seedling-exposure to high temperature, roots with no BRI1 are not short like one would expect - in fact, they are longer as compared to plants with normal BRI1 levels. No correlation between the levels of BRI1 in the plasma membrane and the response of root elongation to high temperature is observed in the *BRI125K* line. Another point of a potential weak correlation is where seedlings treated with BL and *bri1* seedlings have high ratio of root elongation at 26°C over 21°C (Figure 4e and 4g). The authors interpret the results in Figure 4e as root hyposensitivity to BL at 26°C, but it could be also interpreted as hypersensitivity to high temperature. A thorough root morphological analysis for all mutants that show interaction with high temperature, at least as performed in Figure 1 should provide useful information. Sensitivity assays with BL and a BR biosynthesis inhibitor and growth rate analysis will be also informative (also relevance to point 1 below).

Additional comments

1. A high fraction of the experiments in the manuscript studies the role of auxin, concluding that it is not involved. This is fine, but the discussion that revolves around the data occasionally does not sound.

For example:

Line 102-103 "The drop in auxin content is likely counterbalanced by the stabilization of the auxin receptors at higher temperature". This explanation does not sound since at the same time the authors emphasize that their experimental system is different than the one used for studying auxin (rapid temperature change from 22°C to 29°C). In other words, if *tir1afb2* is responsive to 26°C, why the receptor is assumed to be stabilized?

Line 187: "Short-term changes in ambient temperature therefore likely influence root elongation through the rapid modulation of auxin responses, while steroid-dependent root growth control drives later stages of the response to temperature." Auxin promotes hypocotyl response to high temperature also under continuous growth at 28{degree sign}C (Ma et al, PNAS, 2015). Therefore, an alternative explanation could be that auxin is important in response to temperatures above 26{degree sign}C.

To summarize, since auxin is a major subject of this work the authors could examine if *bri1* is hypersensitive to high temperature in a rapid temperature change from 22{degree sign}C to 29{degree sign}C (using the auxin mutants as a positive control) and perform growth rate analysis during the 15 days exposure (as suggested above). This might clarify some of the differences observed between the two hormones or between the two labs.

2. The RNA-seq data was performed by the authors using roots (why 10 days and not 15 days as with the other experiments?). The authors evaluate the common genes that are regulated by BL against root data set (Chaiwanon et al.) but when evaluating the response to auxin, datasets that are not specific for roots were chosen. Could this potentially explain the absence of common genes? It will be more informative if the authors also distinguish the induced and repressed genes when comparing to Chaiwanon et al.

3. Figure 3e: CPD is not among the differential expression of Arabidopsis root genes at 21 and 26{degree sign}C (Table S1), but this gene was analyzed by real time. How many of the BR feedback-regulated genes are included in Table S1? Can the authors confirm more genes?

4. Figure 1c. The root meristem image (lower panel) is unclear.

Reviewer #2 (Remarks to the Author):

In this manuscript, the authors described their identification that under elevated temperature (from 21 to 26 {degree sign}C), the Arabidopsis root growth promotion is not via the activation of auxin signaling pathway. Instead, it is via the down-regulation of brassinosteroid (BR) signaling pathway. More accurately, it is via the degradation of the BR receptor, BRI1. Interestingly, whole genome mRNA-seq analysis indicated that none of the root temperature response genes are shared with auxin response genes. But there is a considerable number of response genes which are shared with BR response genes. Mutants in which auxin signaling pathway is blocked show wide-type like response to the elevated temperature. BR gain-of-function mutant, *bes1-D*, however, showed greatly reduced sensitivity to elevated temperature. The root growth of a *bri1* mutant, on the other hand, showed increased sensitivity to the elevated temperature.

This is a well-written manuscript and easily to follow. But the results really surprised me in a great deal. Because these results are contradictory to our common sense regarding the functions of auxin and BR, as well as the cross-talks between two growth promoting phytohormones. I would like to see more data before I can be convinced that this manuscript is publishable in Nature Communications. Here list some of my major concerns.

1) There are a huge number of publications indicating that auxin and BR share intensive cross-talks. The mRNA-seq analyses provided by the authors showed that elevated temperature share common response genes with BR but not with any one of the auxin response genes. Because these data are really surprising, the authors should reconfirm these results by using different methods such as RT-PCR. At least 10 genes known to be up-regulated by auxin or BR should be included.

2) Seeing the down-regulation of BRI1 protein in higher temperature doesn't always mean the BR signaling has been down-regulated. Otherwise it cannot explain why the roots elongate more at 26 {degree sign}C when BR signaling is down-regulated. It is also contradictory to the fact that *bri1* mutants show shorter root phenotype. It could be caused by the rapid turn-over of BRI1 after the BR signaling has been greatly induced by higher temperature. Reduced root sensitivity to BR, as shown

by a root inhibition assay in Figure 4d, doesn't always mean the BR signaling is reduced. When the BR signaling is highly upregulated could also result in reduced sensitivity. Therefore, more data are needed to clarify these contradictory results.

3) *bes1-D* is not a right mutant to represent the gain-of-function of BR signaling because it has dual specificity. *BRI1* overexpressor and *BAK1* overexpressor should also be included in the assay.

I would like to thank the reviewers for their comments on our manuscript. We addressed most of the comments experimentally and modified the text extensively to strengthen our conclusions and fully convince the reader. Please find below a detailed response to the points raised.

Response to Reviewer #1

- *“The effect of high temperature on BRI1 (Figure 4) is dramatic, by 36 hours no BRI1 protein is detected (extended Data Figure 4). It is therefore puzzling that after 15 days of seedling-exposure to high temperature, roots with no BRI1 are not short like one would expect - in fact, they are longer as compared to plants with normal BRI1 levels. No correlation between the levels of BRI1 in the plasma membrane and the response of root elongation to high temperature is observed in the BRI125KR line.”*

The drop in BRI1 protein accumulation is clearly over-estimated due to confocal imaging. Using higher laser power or gain allows a clear visualization of BRI1 protein accumulating in root cells. The same holds true for western blot analyses where increased exposure allows detection of BRI1 protein at elevated temperature. To avoid misleading the reader, we now provide in Fig. 5c new representative confocal images, with higher gain in the inset to show that BRI1 protein is still detectable.

We show that roots are longer at elevated temperature when plants are subjected to continuous growth at 26°C (compared to 21°C), under long day conditions. As discussed later in this response letter, this observation is not valid under short-term temperature elevation, continuous growth at 29°C under long days, or continuous growth at 26°C under short day conditions. Anyhow, we systematically observe *bri1* root temperature-mediated root growth being greater than what is observed for wild-type plants, and the opposite for BR hypersensitive/constitutive genotypes (*bes1-D*, *bzr1-D* and the triple *bin2/Atsk2-2/Atsk2-3* GSK3 mutant (Fig 4a, b ; Extended Data Fig. 3). This demonstrates genetically that BR signaling negatively regulates root growth at elevated temperature. We performed some additional experiments with plants expressing the non-ubiquitinatable BRI1_{25KR} version and now show that these plants also display a smaller ratio of growth at 26°C/21°C compared to wild-type (Fig. 5e). This smaller response is in the same range than the one observed for *bzr1-D*, or the *bin2/Atsk2-2/Atsk2-3* GSK3 mutant (Extended Data Fig. 3a), and is underestimated since the BRI1_{25KR} transgenic line used harbor slightly lower BRI1 protein level than the control line (Martins et al., 2015). These different observations and their functional consequences for root growth are now extensively discussed throughout the revised version of our manuscript.

- *“Another point of a potential weak correlation is where seedlings treated with BL and *bri1* seedlings have high ratio of root elongation at 26{degree sign}C over 21{degree sign}C (Figure 4e and 4g). The authors interpret the results in Figure 4e as root hyposensitivity to BL at 26{degree sign}C, but it could be also interpreted as hypersensitivity to high temperature. A thorough root morphological analysis for all mutants that show interaction with high temperature, at least as performed in Figure 1 should provide useful information. Sensitivity assays with BL and a BR biosynthesis inhibitor and growth rate analysis will be also informative”*

We provide extensive genetic evidence in our manuscript, using both BR insensitive and hypersensitive/constitutive mutants, that BR signaling negatively impacts on root growth at elevated

temperature (see above). We agree that the hyposensitivity to BL presented Fig. 5e (i.e. high ratio of root elongation at 26°C over 21°C), similar to what is observed in *bri1* is puzzling. We only use the BL root assay to strengthen our observations that BRI1 levels are down at 26°C. Conclusions on the involvement of BR signaling in root growth responses at elevated temperature are solely drawn from genetic proofs presented in this study, rather than on exogenous application of BL that will i) provide non-physiological concentration of ligand, ii) will abolish spatial and temporal control of BR responses that we know are of high importance (Work from Sigal Savaldi-Goldstein's lab and Zhi Yong Wang's lab), and iii) will also activate other BR receptors such as BRL1 and BRL3 for which we have not demonstrated regulation by elevated temperatures.

Although we formally agree with Reviewer 1 on the conclusion regarding the hyposensitivity to BL vs hypersensitivity to temperature presented in Fig. 5e (former Fig. 4e), we stand by our conclusion since we also demonstrate in the manuscript that under elevated temperature the levels of the BR receptor BRI1 are down (Figure 5a, c), and BR signaling is less active (Figure 4c, d, e). Taken together, these several observations clearly argue for a hyposensitivity to BRs. Therefore, we feel the proposed experiments aiming at characterizing in detail the primary root of *bri1* and *bes1-D* mutants (by measuring meristem length, size of elongated cells, cell divisions, etc.) treated with BRs and the BR inhibitor BRZ at both temperature, which are quite tedious and time-consuming, will not provide further meaningful information regarding the role of BR signaling in elevated temperature. We decided not to do these experiments for the sake of time to rather focus on figuring out the discrepancy about the role of auxin between our work and the work from Wang et al (2016) (see below).

- *"A high fraction of the experiments in the manuscript studies the role of auxin, concluding that it is not involved. This is fine, but the discussion that revolves around the data occasionally does not sound. For example: Line 102-103 "The drop in auxin content is likely counterbalanced by the stabilization of the auxin receptors at higher temperature". This explanation does not sound since at the same time the authors emphasize that their experimental system is different than the one used for studying auxin (rapid temperature change from 22{degree sign}C to 29{degree sign}C)."*

We agree with reviewer 1. We now provide in Supplementary Fig. 6b and 6c evidence that TIR1 is also stabilized under our experimental setup (i.e. continuous growth at 21°C or 26°C under long day conditions).

- *"Line 187: "Short-term changes in ambient temperature therefore likely influence root elongation through the rapid modulation of auxin responses, while steroid-dependent root growth control drives later stages of the response to temperature." Auxin promotes hypocotyl response to high temperature also under continuous growth at 28{degree sign}C (Ma et al, PNAS, 2015). Therefore, an alternative explanation could be that auxin is important in response to temperatures above 26{degree sign}C. Since auxin is a major subject of this work the authors could examine if *bri1* is hypersensitive to high temperature in a rapid temperature change from 22{degree sign}C to 29{degree sign}C (using the auxin mutants as a positive control) and perform growth rate analysis during the 15 days exposure (as suggested above). This might clarify some of the differences observed between the two hormones or between the two labs."*

We speculated that auxin may drive responses to short-term changes in temperature while BR would be more important for later phases of the response, like already postulated for hypocotyls (Stavang

et al., 2009), reviewer 2 argues that in the case of hypocotyls auxin also mediate responses to continuous growth at elevated temperature (Ma et al., 2016) and that auxin may be required for temperatures above 26°C.

To better understand the discrepancy between our data and the ones from Wang et al (2016), we decided to perform several experiments testing the several parameters that differ between the two studies. These differences are : i) we use continuous growth under elevated temperature, since the rationale of our study is to assess the impact of global warming and anticipate on its consequences on plant growth, while the other group uses short-term exposure to warm conditions ; ii) we use 21-26°C versus 22/29°C for Wang et al (2016) ; iii) we use long days whereas short day conditions were used by Wang et al. We now provide genetic evidence that BR mutants show differential root growth at elevated temperature compared to wild type, regardless of photoperiod (long vs short days), the duration of warm conditions (continuous vs short-term), and the severity of temperature elevation (21/26°C or 22-29°C ; Supplementary Fig. 3b, c, d). In these exact same conditions, the *tir1* and *tir1afb2* showed wild-type responses. At present, we do not know what other factor may change the relative contribution of BR vs auxin in root temperature responses and explain the discrepancy between work done in the two labs, but we feel we have given our best to demonstrate the global role of BR signaling in root growth upon warmth. This is now extensively discussed in the revised version of the manuscript.

- *“The RNA-seq data was performed by the authors using roots (why 10 days and not 15 days as with the other experiments?). The authors evaluate the common genes that are regulated by BR against root data set (Chaiwanon et al.) but when evaluating the response to auxin, datasets that are not specific for roots were chosen. Could this potentially explain the absence of common genes? It will be more informative if the authors also distinguish the induced and repressed genes when comparing to Chaiwanon et al.”*

We apologize for this mistake. RNAseq comparison was indeed made with seedling-derived RNAseq auxin datasets where root-expressed genes are under-represented due to the greater mass of shoots compared to roots. Comparison of our temperature dataset with the root auxin dataset from Chaiwanon et al (2015) shows overlaps with auxin-regulated genes. When comparing temperature-regulated genes and genes specifically regulated by BRs or specifically regulated by auxin, we noticed that the overlap was greater with BR-specific genes. However, the revised version of the manuscript is spun in a radically different way and we do not use this observation anymore.

We chose to perform the RNAseq experiment at day 10 for two reasons. First, plants were continuously grown under elevated temperature, indicating that the altered root length observed at day 15 is the consequence of changes in gene expression occurring from germination. The phenotypic outcome measured at day 15 reflects gene expression occurring in the preceding days/hours. More importantly, although not discussed in the manuscript, we also observe differences in lateral root emergence at 21 and 26°C. Performing RNAseq at day 10 allow us to to limit the indirect effects of different root system architecture on gene expression.

- *“Figure 3e: CPD is not among the differential expression of Arabidopsis root genes at 21 and 26{degree sign}C (Table S1), but this gene was analyzed by real time. How many of the BR feedback-regulated genes are included in Table S1? Can the authors confirm more genes?”*

Reviewer 1 is right. CPD is not in Table S1 since it did not pass the 1.3-fold cutoff in the RNAseq experiment. Genomic response to exogenous application of BL are notorious for the low amplitude in induction or repression (Goda et al., 2002 ; Mussig et al., 2002). Confirmation by qRT-PCR however showed that CPD is indeed upregulated upon elevated ambient temperature. To strengthen our conclusions, we now provide a total of 4 BR homeostasis genes known to be regulated by BRs and that came up as regulated by elevated temperature in our study. The BR feedback-regulated genes CPD and STE1/DWF7 both show increased expression upon warm conditions (Fig. 4e), and the BR metabolism genes BAS1 and CYP72C1/SOB7 that are induced upon BL exposure are both downregulated by elevated temperature (Fig. 4f). Expression of these 4 genes confirms that BR signaling is downregulated upon elevated temperature.

- *“Figure 1c. The root meristem image (lower panel) is unclear”*

We provided new images to better visualize meristematic cells.

Response to Reviewer #2

- *“There are a huge number of publications indicating that auxin and BR share intensive cross-talks. The mRNA-seq analyses provided by the authors showed that elevated temperature share common response genes with BR but not with any one of the auxin response genes. Because these data are really surprising, the authors should reconfirm these results by using different methods such as RT-PCR. At least 10 genes known to be up-regulated by auxin or BR should be included.”*

Comparison of temperature-regulated genes with auxin-regulated genes was originally performed using auxin-regulated gene list generated from whole seedlings where root-expressed genes are under-represented due to the greater mass of shoots compared to roots (see response to reviewer 1). This largely explains why no overlap whatsoever was found. In addition, we agree with reviewer 2 that auxin and BRs are notorious for sharing many target genes, but this has mostly been studied in the context of hypocotyls. The situation in roots is more elusive and recent work from Chaiwanon et al (2015) nicely showed that BR and auxin genomic responses actually antagonistically regulate gene expression in the root.

Considering that we decided to change the way we tell the whole story in the revised manuscript, we do not make use of this observation anymore and decided not to follow up on the expression of BR:auxin co-regulated root genes.

- *“Seeing the down-regulation of BRI1 protein in higher temperature doesn't always mean the BR signaling has been down-regulated. Otherwise it cannot explain why the roots elongate more at 26 {degree sign}C when BR signaling is down-regulated. It is also contradictory to the fact that bri1 mutants show shorter root phenotype. It could be caused by the rapid turn-over of BRI1 after the BR signaling has been greatly induced by higher temperature. Reduced root sensitivity to BR, as shown*

by a root inhibition assay in Figure 4d, doesn't always mean the BR signaling is reduced. When the BR signaling is highly upregulated could also result in reduced sensitivity. Therefore, more data are needed to clarify these contradictory results."

Plants grown at elevated temperature possess lower BRI1 levels (see response to point 1 raised by reviewer 1, and Fig. 5c, inset), but nothing compared to a full knock-out plant like the *bri1* T-DNA insertion mutant used in this study. Therefore, it is hard to compare *bri1* with plants grown at higher temperature. Regardless, we used established markers of BR signaling such as the phosphorylation status of BES1, or expression of BR metabolism genes that are regulated by BRs, to show that BR signaling is downregulated at elevated temperature.

BRs have mostly known for their growth promoting effect on light-grown hypocotyls. However, in roots, BRs promote growth at very low concentrations and inhibit growth at higher concentration. Therefore, a downregulation of BR signaling may yield longer roots if BR levels are supraoptimal. We tested this hypothesis by performing a BL dose-response analysis on plants grown at 26°C, but failed to see any decrease in root length at very low dose of BL (Supplementary Fig. 7), indicating that endogenous BR levels are rather optimal than supraoptimal. Considering that insensitive and hypersensitive/constitutive BR mutants show greater and weaker ratio of root length at elevated versus standard temperature, respectively (Fig. 4b and Supplementary Fig. 3b, c, d), we think that BR signaling is wired differently and impinges on a downstream signaling pathway specifically at elevated temperature to promote root elongation upon warmth. We ruled out the role of auxin in our conditions. Future work will address the role of other hormones such as ABA, ethylene or gibberellins in root responses to elevated temperature and their relationship with BR signaling. This is now extensively discussed in the revised version of the manuscript.

- "bes1-D is not a right mutant to represent the gain-of-function of BR signaling because it has dual specificity. BRI1 overexpressor and BAK1 overexpressor should also be included in the assay."

We agree with reviewer 2 and now provide additional genetic evidence that BR signaling negatively regulates root growth promotion at elevated temperatures, including the use of *bzr1-D*, the triple *bin2/Atsk2-2/Atsk2-3* GSK3 mutant that is impaired in steps upstream in the BR signaling pathway (Supplementary Fig. 3a). Besides, we demonstrate that the non-ubiquitinatable BRI1_{25KR}-expressing plants, which display BR hypersensitivity phenotypes (Martins et al., 2015) and are resistant to the temperature effect (Fig. 5f), are defective in temperature-mediated root elongation. The post-transcriptional control of BRI1 by temperature prevents us from using 35S::BRI1 suggested by reviewer 2, since still sensitive to temperature (Supplementary Fig. 4b). In addition and to our knowledge, BAK1 overexpressors are only available as tagged versions that are partly inactive, especially for PAMP-triggered immunity (PTI) (Ntoukakis et al., 2011). Since the top up-regulated genes upon elevated temperature are related to PTI and that activation of defense usually impacts on growth, we felt the use of BAK1 overexpressors was rather dangerous and certainly not better than using *bin2/Atsk2-2/Atsk2-3*.

Reviewers' comments:

Reviewer #1 (Remarks to the Author):

Review:

The revised m/s largely addressed my concerns. Few unclear points remained and I believe they could be addressed by modifying the text as applied:

1. In the rebuttal letter, the authors consider the genetic data as a reliable evidence for BR inhibition of root elongation in response to high temperature, unlike the use of exogenous hormone. While I agree with the authors that the use of exogenous BL could be non physiological etc, the data in Fig. 5d,e remained confusing. As such, I suggest that the authors also briefly explain this point in the discussion, i.e., why high BL levels enhance root response to high temp.

2. Short day experiment:

Line 197: " BR signaling dampens root growth.... independently of the photoperiod..." (also discussed in lines 309-311).

This result/conclusion is not fully consistent with the model proposed, where high temperature reduces BRI1 levels and therefore root elongates. In short days wild-type roots are inhibited (is this significant? root length in each condition is not shown).

Also, if in short days wild type roots are inhibited - the stronger response of bes1-D and the lack of response of bri1 (the elevated ratio of bri1 vs wild-type is not significant, right?) is a genetic evidence that BR signaling positively (not negatively) regulates root responses to ambient temperature under this photoperiod condition.

These points should be better discussed/clarified.

Minor points:

SD is missing from some bars, see Supplementary Figure 3b, Fig. 4e

Some typos:

Line 62 - In

Line 145: "Steroid signaling IS involved?"

212-214 - Figure 4 (not 3)

Supp., line 47 - TI1-Venus , should be TIR1

Reviewer #3 (Remarks to the Author):

The present study reported a novel discovery that the elevated temperature induced root growth by inhibiting steroid hormone signaling. The discovery appears a little out of the expectation as the previous study has suggested auxin is the major player in the process in the shoot/hypocotyl tissues. It's also a surprise considering brassinosteroids were generally thought as a class of growth promoting hormones. Thus, I read the manuscript with high interests and believe that the conclusion of the study has great significance for enhancing our comprehension of the hormone functions in a distinct biological process.

On the other hand, however, considering the potential impact of the study, I have a little concern on the suggested role of brassinosteroid as a repressor of the root growth, as mentioned and discussed

by the authors. The short root phenotype of *bri1* mutant at both high and low temperature (Fig. 4a) seems inconsistent with the model that heat induced BRI1 degradation to inhibit root growth. Despite the authors' conjecture that BR signaling may affect an additional component/pathway to inhibit the root growth, perhaps there are other possibilities to explain the phenomenon, which may need to be discussed. For example, as shown by authors, heat may down-regulate BR to inhibit cell division but promote cell elongation, and *bri1* mutant might have a dominant defection on cell division thus showing short root. To confirm this, the analysis of *bri1* mutant root under normal condition might be required. In addition, according to the authors' data, heat may directly induce BR accumulation (might not be caused by BRI1 inhibition) to induce root elongation. To exclude this possibility, BR synthetic mutants as controls should be included for the physiological analysis. Also, the root growth of *bri1* mutant as well as other BR-related mutants to exogenous BR application should be tested to ensure these materials are reliable. The expression of BR synthetic genes should also be evaluated in the root of *bri1* mutant. This kind of taken-for-granted information will make the story more convincing considering the conclusion is a little surprise. Regarding the idea that BR affects other phytohormones to regulate the heat-induced cell elongation, the authors may refer to the recent papers showing BR targets GA synthesis to regulate cell elongation (Unterholzner et al, *Plant Cell*, 2015; Tong et al, *Plant Cell*, 2014), especially in rice showing that that BR inhibits cell elongation by inducing gibberellin inactivation in both shoot and root (Tong et al, *Plant Cell*, 2014). In addition, it was shown SHOEBOX targeted GA synthetic gene to regulate root meristem activity (Li et al, *PLoS Genetics*, 2015), and recently it was shown SHOEBOX/SMOS1/RLA1 (same protein with different names) interacts with both OsBZR1 and DLT, two critical rice BR signaling transcriptional factors, to regulate BR responses (Hirano et al., *Mol Plant*, 2017; Qiao et al., *Plant Cell*, 2017). I suggest the authors to appropriately cite the closely related articles to support the proposed idea that BR affects additional phytohormone pathway to regulate root growth. Of course, it would be much better if the authors can provide some direct evidence to support the idea.

We would like to thank both reviewers for the comments and suggestions to strengthen the conclusions drawn in our manuscript. Below is a detailed response to all points raised :

Reviewer #1

1. *"In the rebuttal letter, the authors consider the genetic data as a reliable evidence for BR inhibition of root elongation in response to high temperature, unlike the use of exogenous hormone. While I agree with the authors that the use of exogenous BL could be non-physiological etc, the data in Fig. 5d,e remained confusing. As such, I suggest that the authors also briefly explain this point in the discussion, i.e., why high BL levels enhance root response to high temp."*

As suggested by Reviewer #1, we now provide explanation in the discussion section about the discrepancy between conclusions drawn from the many genetic data, and what has been observed for exogenous application of BRs.

This part reads : "Our work provides several complementary evidence using genetics, genomics, biochemical and cell biology approaches that elevated temperature impinges on BR signaling at the level of BRI1 to control steroid signaling-dependent root elongation. Although genetic analyses clearly point to negative role of BR signaling in root responses to elevated temperature in long days, our photoperiod of reference in this study, exogenous application of BL is associated with an enhanced response to temperature under the same conditions. Such discrepancy may be explained by the non-physiological concentration of exogenously applied ligand, which not only abolishes spatial and temporal control of BR responses that are crucial for proper root growth^{21, 22, 23, 26, 38}, but also likely activates other BR receptors such as BRL1 and BRL3 for which temperature regulation has not been demonstrated."

2. *"Short day experiment: Line 197: " BR signaling dampens root growth... independently of the photoperiod..." (also discussed in lines 309-311). This result/conclusion is not fully consistent with the model proposed, where high temperature reduces BRI1 levels and therefore root elongates. In short days wild-type roots are inhibited (is this significant? root length in each condition is not shown). Also, if in short days wild type roots are inhibited - the stronger response of bes1-D and the lack of response of bri1 (the elevated ratio of bri1 vs wild-type is not significant, right?) is a genetic evidence that BR signaling positively (not negatively) regulates root responses to ambient temperature under this photoperiod condition. These points should be better discussed/clarified."*

We agree with Reviewer #1 that the genetic contribution of BR signaling to root growth varies with the photoperiod. BR signaling negatively regulate root responses to elevated temperature in long days, while promoting root responses to temperature in short days. Although the underlying mechanisms are unknown, and clearly belong to a follow up story, we now clearly discuss this point (see below). It would indeed be interesting to determine how short day impact on BR signaling and BRI1 levels.

In short days, wild-type roots are significantly shorter at elevated temperature. We decided not to present the actual root length (since very space consuming) but to stick with the ratio 26/21 not to confuse the reader and to save this new aspect of the story for future work.

The corresponding paragraph has been added to the result section : "Overall, constant growth at elevated temperature has a profound influence on plant root growth depending on the photoperiod, with roots elongating more under long days and roots becoming shorter under short days. Our results indicate that BR signaling negatively regulates root responses to temperature elevation under long days, while positively regulating temperature-mediated root elongation under short days.

Regardless, the primary root length ratio from plants grown at 26°C over 21°C are always greater for *bri1* and lower for *bes1-D* (Fig. 4b, Supplementary Fig. 4b-d)."

We also discuss this point in the discussion section, in the light of the crosstalk between BRs and light mediated by the BZR1/PIF module : "The photoperiod strongly influences root growth and the role of BR signaling in root responses to elevated ambient temperature. While we have clearly demonstrated that warm conditions impact BR signaling by decreasing BRI1 levels in long days, the influence of short day conditions on BRI1 levels and BR signaling will have to be investigated. The PIF-BES1/BZR1 module, which integrates BRs, temperature and light inputs in hypocotyls^{48, 49}, may also contribute to root temperature responses and convey photoperiod input. It is therefore conceivable that temperature also crosstalks with BR and light signaling at the levels of BES1/BZR1 through interaction with a yet to be identified root-expressed PIF, providing more complexity in the mechanisms of ambient temperature control through BR signaling. We genetically excluded the contribution of PIF4, but other PIFs have already been reported to participate in root growth⁵⁰ and may therefore contribute to the integrated root growth control by BRs and temperature."

Minor points:

- "SD is missing from some bars, see Supplementary Figure 3b, Fig. 4e"

The error bars were actually not missing but were very small for one sample, and therefore barely visible at regular magnification. We have artificially increased the size of the error bar so it is visible to the reader.

- "Some typos:

Line 62 – In

Line 145: "Steroid signaling IS involved?"

212-214 – Figure 4 (not 3)

Supp., line 47 - T11-Venus , should be TIR1"

All the typos have been corrected

Reviewer #3

We are grateful to Reviewer #3 for joining the evaluation process of our manuscript, and for insightful comments.

1. "The short root phenotype of *bri1* mutant at both high and low temperature (Fig. 4a) seems inconsistent with the model that heat induced BRI1 degradation to inhibit root growth. Despite the authors' conjecture that BR signaling may affect an additional component/pathway to inhibit the root growth, perhaps there are other possibilities to explain the phenomenon, which may need to be discussed. For example, as shown by authors, heat may down-regulate BR to inhibit cell division but promote cell elongation, and *bri1* mutant might have a dominant defection on cell division thus

showing short root. To confirm this, the analysis of bri1 mutant root under normal condition might be required.”

We fully agree with reviewer #3 about the possibility of other scenarios to explain the observed results. Heat appears to downregulate BR signaling to inhibit cell division in the root meristem and promote cell expansion. It is therefore conceivable that *bri1* is mostly affected in cell division, explaining the *bri1* short root phenotype under standard conditions, while retaining the ability to promote cell elongation upon warmth. This would however hardly explain why *bri1* over-responds to temperature elevation compared to wild-type. Nevertheless and as suggested by Review #3, we investigated the root growth parameters of *bri1* mutant and observed strong reduction in meristem size and cell expansion. This is now shown in Supplementary Fig. 3c, d. These observations confirm previous results from Gonzalez-Garcia et al, Development 2011 and Hacham et al., Development 2011.

We also monitored *bri1* root parameters at elevated temperatures to evaluate the basis of over-responsiveness to 26°C (Supplementary Fig. 3c, d), and observed that heat triggers cell proliferation and expansion, in contrast to WT where meristem size is reduced by warmth. This suggests that typical BR signaling and responses are likely not the sole driver of temperature-mediated growth, but that an additional cue dependent on BR signaling reduction unleashed growth at elevated temperature.

All this is now extensively discussed in the manuscript, as requested and reads :” The fact that root elongation in responses to continuous warm conditions and long day photoperiod is mediated by a decrease in BRI1 levels and attenuated BR signaling is therefore puzzling. In roots, BRs promote growth at very low concentrations and inhibit growth at higher levels, under standard growth temperature ²². BR concentration may however be supraoptimal for root growth at 26°C. Consequently, the temperature-dependent downregulation of BR signaling would yield longer roots. Two lines of evidence argue against this hypothesis. First, the *bri1* mutant is still responsive to temperature elevation. Second, dose-response analyses to exogenously applied BL failed to confirm supraoptimal root BR levels since root growth was not inhibited at very low BR concentrations at 26°C (Supplementary Fig. 9). A second explanation to the BR-mediated elongation upon heat may rely in the relative contribution of BRI1 and BR signaling to cell division and expansion. *bri1* may mostly be affected in cell division under standard temperature, explaining the *bri1* short root phenotype, while retaining the ability to promote cell elongation upon warmth. This scenario is however not supported by our observations that *bri1* is severely impaired in both cell proliferation and elongation, and cannot easily explain why *bri1* over-responds to temperature elevation compared to wild-type. Alternatively, BR signaling may impinge on a downstream signaling pathway, specifically at elevated temperature, to differentially regulate root cell division and expansion. Downregulation of BRI1 by heat, or all the more the total absence of BRI1, would unleash such downstream pathway and yield longer roots. This would explain the greater ratio of root length at elevated over standard temperature for *bri1* compared to wild-type plants, regardless of the duration or the photoperiod, and the decreased ratio for *bes1-D*. We ruled out the involvement of auxin and TIR1/AFB2 in our growth conditions. Elevated temperature therefore likely influences root growth *via* another plant hormone, under the control of BR signaling.”

2. “In addition, according to the authors’ data, heat may directly induce BR accumulation (might not be caused by BRI1 inhibition) to induce root elongation. To exclude this possibility, BR synthetic mutants as controls should be included for the physiological analysis.”

As suggested by Reviewer #3, we monitored root responses to elevated temperature of two well-established BR-deficient mutants : *det2* and *dwf4*. If a possible increase in BR levels was causing the enhanced root elongation at elevated temperature observed in wild-type plants, both BR-deficient

mutants should be impaired in such response. As now shown supplementary Fig. 3, *det2* and *dwf4* showed slightly shorter roots at 21°C, but responded as wild-type plants to 26°C conditions. This rules out the possibility that increased BR synthesis drives the further elongation observed. However, an interesting observation arose from these new experiments since BR-deficient mutants responded like wild-type while all BR signaling mutant tested were affected. We added a short paragraph in the discussion section to explain such observation. The paragraph reads : “Similarly, the fact that BR-biosynthetic mutants respond like wild-type to warm conditions while all insensitive or hypersensitive/constitutive BR genotypes tested respond differentially is surprising but likely reveals some specificity in the cell type or developmental stage where BR are synthesized, perceived by BRI1 and BRLs and where temperature acts on BR signaling.”

3. *“Also, the root growth of bri1 mutant as well as other BR-related mutants to exogenous BR application should be tested to ensure these materials are reliable. The expression of BR synthetic genes should also be evaluated in the root of bri1 mutant. This kind of taken-for-granted information will make the story more convincing considering the conclusion is a little surprise.”*

To ascertain that the materials we used are reliable, we tested the response of all BR-related mutants tested to BRs or BRZ, and the auxin mutants to IAA. All mutants behaved as expected, further strengthening our observations (supplementary Fig. 10). In addition, the mRNA accumulation of BR-regulated genes was investigated as suggested in wild-type and *bri1*, to ensure that they behaved like expected (i.e. BR feedback-regulated genes upregulated in *bri1*, and BR-induced genes downregulated in *bri1*) (supplementary Fig. 5) so our gene expression data is more convincing.

4. *“Regarding the idea that BR affects other phytohormones to regulate the heat-induced cell elongation, the authors may refer to the recent papers showing BR targets GA synthesis to regulate cell elongation (Unterholzner et al, Plant Cell, 2015; Tong et al, Plant Cell, 2014), especially in rice showing that that BR inhibits cell elongation by inducing gibberellin inactivation in both shoot and root (Tong et al, Plant Cell,2014). In addition, it was shown SHOEBOX targeted GA synthetic gene to regulate root meristem activity (Li et al, PLoS Genetics, 2015), and recently it was shown SHOEBOX/SMOS1/RLA1 (same protein with different names) interacts with both OsBZR1 and DLT, two critical rice BR signaling transcriptional factors, to regulate BR responses (Hirano et al., Mol Plant, 2017; Qiao et al., Plant Cell, 2017). I suggest the authors to appropriately cite the closely related articles to support the proposed idea that BR affects additional phytohormone pathway to regulate root growth. Of course, it would be much better if the authors can provide some direct evidence to support the idea.”*

We thank Reviewer #3 for pointing these references on the link between BRs and GA in cell division/elongation. We now cite in our revised version of the manuscript these references, as well as others, to support the idea that BRs is connected to other hormonal signal pathways. We will start to experimentally address the relevance of the BR/GA crosstalk in the future, but feel this does not belong to the present manuscript, so we did not add any experiment related to this specific point but only discussed this, as suggested by the reviewer.

The discussion now reads : “Gibberellins (GA) are known to modulate both the rate of cell proliferation and the extent of dividing and post-mitotic cell expansion^{39, 40, 41}. Several recent reports in Arabidopsis and rice established a link between BR signaling to GA metabolism or to GA signaling in the control of plant growth^{42, 43, 44, 45, 46}. Although in most cases, BRs positively regulate GA to boost growth, more complex interactions between the two hormones have been reported. BRs were notably found in rice roots to negatively regulate GA⁴⁷. BRs indeed promote the expression of the GA2ox3 GA-inactivating gene and inhibit the GA20ox3 GA biosynthetic gene, leading to increased

abundance of the central GA DELLA repressor SLR1⁴⁷. Decreased BR signaling has therefore the ability to lead to more bioactive GA levels and enhanced GA signaling under certain circumstances or cell types.”

REVIEWERS' COMMENTS:

Reviewer #3 (Remarks to the Author):

I appreciate that the authors took the reviewer's concerns seriously and address the most questions raised previously. As also mentioned by the authors, it is apparent that specific functions of BRs are largely dependent on the species, tissues, growth conditions, cell types and so on. I look forward to seeing the follow-up studies by the authors regarding the BR functions in root in balancing the growth requirements and the responses to growth conditions. At present, I thought the revised version of the manuscript provides evidence strong enough to support the major finding that the steroid hormones or the BRI1 receptor is involved in warm-induced root elongation responses.